# Peptidyl Arginine Deiminases in Chronic Diseases: A Focus on Rheumatoid Arthritis and Interstitial Lung Disease

**DOI:** 10.3390/cells12242829

**Published:** 2023-12-13

**Authors:** Karol J. Nava-Quiroz, Luis A. López-Flores, Gloria Pérez-Rubio, Jorge Rojas-Serrano, Ramcés Falfán-Valencia

**Affiliations:** 1HLA Laboratory, Instituto Nacional de Enfermedades Respiratorias Ismael Cosío Villegas, Tlalpan, Mexico City 14080, Mexico; krolnava@hotmail.com (K.J.N.-Q.); glofos@yahoo.com.mx (G.P.-R.); 2Programa de Doctorado en Ciencias Médicas Odontológicas y de la Salud, Universidad Nacional Autónoma de México (UNAM), Mexico City 04510, Mexico; 3Rheumatology Clinic, Instituto Nacional de Enfermedades Respiratorias Ismael Cosío Villegas, Tlalpan, Mexico City 14080, Mexico

**Keywords:** peptidyl arginine deiminases, PAD, citrullination, rheumatoid arthritis, interstitial lung disease, inflammation, extracellular matrix remodeling

## Abstract

Protein citrullination is accomplished by a broad enzyme family named Peptidyl Arginine Deiminases (PADs), which makes this post-translational modification in many proteins that perform physiological and pathologic mechanisms in the body. Due to these modifications, citrullination has become a significant topic in the study of pathological processes. It has been related to some chronic and autoimmune diseases, including rheumatoid arthritis (RA), interstitial lung diseases (ILD), multiple sclerosis (MS), and certain types of cancer, among others. Antibody production against different targets, including filaggrin, vimentin, and collagen, results in an immune response if they are citrullinated, which triggers a continuous inflammatory process characteristic of autoimmune and certain chronic diseases. PAD coding genes (*PADI1* to *PADI4* and *PADI6*) harbor variations that can be important in these enzymes’ folding, activity, function, and half-life. However, few studies have considered these genetic factors in the context of chronic diseases. Exploring PAD pathways and their role in autoimmune and chronic diseases is a major topic in developing new pharmacological targets and valuable biomarkers to improve diagnosis and prevention. The present review addresses and highlights genetic, molecular, biochemical, and physiopathological factors where PAD enzymes perform a major role in autoimmune and chronic diseases.

## 1. Introduction

Protein citrullination is accomplished by a large enzyme family named Peptidyl Arginine Deiminases (PAD), which makes this post-translational modification in a large number of proteins that perform ordinary and pathogenic mechanisms in the body, from keratinocyte differentiation to the involvement of signaling pathway and/or estrogen regulation, to name a few examples.

The role of PAD in the pathophysiological mechanism of diseases such as autoimmune or chronic diseases is complex, including involvement in the production of antibody target molecules (autoantibodies in autoimmune diseases), which could be a consequence of abnormal modification by PAD or alteration of activity in joints as well as in other organs and systems. A change in the activity of these enzyme-functional proteins is mainly associated with an inflammatory process that triggers an exaggerated response.

This review aims to investigate PAD enzymes, which significantly affect autoimmune and chronic diseases. This modification has been implicated in various physiological and pathological processes, including chronic inflammatory diseases such as rheumatoid arthritis and lung diseases related to autoimmune and chronic diseases. We will focus on the PAD enzymes involved in rheumatoid arthritis (RA) and pulmonary diseases.

## 2. PAD Enzyme

### 2.1. Peptidyl Arginine Deiminases “PADs”

Post-translational modifications (PTM) include genetic expression, enzymatic activity, and protein stability, critical processes in cell biology. These PTMs can be made by phosphorylation, acetylation, methylation, and citrullination. This last one is catalyzed by a calcium (Ca^2+^)-dependent enzyme family called peptidyl arginine deiminases (PADs; EC 3.5.3.15) [1,2].

### 2.2. Biological Function

PADs regulate many processes in fundamental cellular mechanisms; there are five isoenzymes in mammals, PAD1 to PAD4 and PAD6, which have 70–95% of homology in the amino-acid sequence [3] and different substrates and specific distribution/expression in the body [1,4], establishing an addressed biological function. Some biological functions and distribution in the body of PADs are shown in Table 1.

The most described function is keratin citrullination, which is a necessary process for the cornification of the epidermis performed by PAD1 [13]; PAD2 has a major role in astrocyte signaling on the central nervous system (CNS) [13]; PAD3 has been associated with the loss of regenerative ability in neural cells [14]; PAD4 is mainly located in cell nucleus, and among its functions is to regulate chromatin decondensation by histone citrullination, gene regulation, and neutrophil extracellular trap formation [15]; finally, PAD6 is involved in oocyte formation and growth, and microtubule regulation and movement [13].

### 2.3. Enzyme Mechanism

Protein citrullination mediated by PAD is conducted by hydrolysis in the guanidinium bond of peptidyl arginine producing peptidyl citrulline and ammonium, which contains a large number of guanidinium modifications; these enzymes are named aminotransferases [16]. PADs have specific inhibitors determined by chemical reactions. In PAD2, guanidinium neutralization positive charge is replaced by urea, decreasing cysteine nucleophilic p*K*_a_, compared with the reverse protonation mechanism in PAD1, 3, and 4 [1].

At the protein level, citrullination induces molecular mass reduction to 1.0 Da by each modified arginine, losing its positive charge [3,17]. The isoelectric point provided by the hydrogen bond allows interaction with other proteins [2,10,18]. The negative net charge is believed to recognize the substrate of arginine residues with a positive charge and the union of essential calcium ions [19].

Citrullinated neoepitope production in extracellular space makes them reachable to the immune system and then initiates an immune response, which modifies citrullinated protein function and its half-life [20].

### 2.4. Enzyme Regulation

Specific biochemical reactions mediate the PAD regulation process. Ca^2+^ ion is the most described regulator. Low intracellular Ca^2+^ concentrations make PAD inactive [3]. However, when there is an alteration (triggered by apoptosis, necrosis, or oxidative stress), Ca^2+^ levels increase and activate the enzyme [3,7,21,22], which makes the citrullination unspecific, reducing the positive charges and stability of the nucleosome as illustrated in Figure 1A [23].

PAD enzymatic activity is also regulated by pH, exhibiting a maximum peak between 7.0 and 8.0 on PAD2 and PAD4 in the ionization process [24]. Although PAD has the same covalent catalysis mechanism by a cysteine residue in its active site, they show differences in kinetic properties [25,26,27].

PAD substrates affect enzymatic regulation. The most reported are estradiol, filaggrin, vimentin, myelin basic protein (MBP), fibrinogen, some chemokines, and histones [17,28,29]. Estrogens regulate PAD4 expression by improving promoter binding sites of *PADI4* [2], and nuclear substrate for PAD4 are histones H2A, H3, and H4 [25]. 

Figure 1B illustrates the physiological mechanism that generates antigens, potentially initiating an autoimmune, inflammatory response leading to relevant diseases, as described in Section 4. Modifications within the gene structure of each isoform were observed to alter the expression of PAD proteins, as detailed below.

## 3. Genes

### 3.1. Localization and Structure of the Gene

PAD coding genes are located in chromosomal region 1p35-36, with a ~355Kb length; they are named *PADI* genes [25,30]. However, as shown in Figure 2, each gene has a different length and structure.

### 3.2. Genetic Variants

Many genetic variants have been described in coding regions (exons), non-coding regions (introns), and nearby regions. These genetic variants are classified as single-nucleotide polymorphisms (SNPs), insertions, deletions, etc., [31,32,33], and details are shown in Table 2.

Among the genetic variants reported in *PADI* genes, SNPs represent ~90% of the total variation, of which ~50% are located in structural/regulatory regions (srSNP: 3′UTR, 5′UTR, and intronic), and only 3–8% are in coding regions (cSNP) that can be either synonymous or non-synonymous [34,35]. These variants can affect the proteins by modifying their structure, function, electric charge, or substrate affinity. Due to the many SNPs located in different regions in *PADI* genes and the effect of their biological consequence, they have been evaluated in genetic association studies with complex traits.

## 4. Citrullination, PAD, and Physiopathology

Citrullination causes structural and functional changes in their biological targets (irreversible modification), such as a loss of positive charge, conformational modification, alterations in protease, or interaction protein affinity (Figure 1B). This physiological process may generate new antigens that induce a systemic response [13].

Due to these modifications, citrullination has become a major topic in the study of pathological processes. It has been related to some chronic and autoimmune diseases, including RA, chronic lung diseases, multiple sclerosis (MS), Alzheimer’s disease, and certain types of cancer [1,4,18,36].

Multiple sclerosis (MS) is a severe autoimmune and complex disease affecting electrical neural conduction in the CNS by gradually losing myelin in oligodendroglial cells. It has been proposed that MBP hypercitrullination plays a significant role in this disease [3]. MBP is a component in myelin sheaths that naturally protects neural axons and is synthesized by oligodendrocytes; MBP deimination can alter proteolysis sensibility by unsettling CNS cells due to an altered interaction of myelin sheaths phospholipids [37,38]. PAD4 has increased levels and activity in MS patients caused by either a demyelination process or a dysfunction in myelin repairment; either way, oligodendrocyte survival is affected [39].

An increase in PAD2 and citrullinated proteins like vimentin have been detected in some neurodegenerative diseases like Alzheimer’s [13,40]. Additionally, PAD3 has been related to decreased regenerative ability in neural cells [41,42]. It has been addressed in other neurodegenerative diseases like schizophrenia; however, it has not presented meaningful results associated with *PADI2* [43].

Some types of cancer have been linked to altered roles of PAD and their coding genes. It has been found that tumors, including squamous cell esophageal carcinoma, have an abnormally increased expression in *PADI4*. Furthermore, this increase has been associated with carcinogenesis, progress, and metastasis in other tumors [44]. However, expression decreases when the tumor is extirpated [45]. It has been argued that tumors can induce autoantibodies for assembling after new epitope recognition [46].

In breast cancer, PAD2 has a major role in cell proliferation [47], and *PADI2* is over-expressed in luminal subtype cells [48]. *PADI4* has been related to regulating gene expression, transcriptional activation, and interaction with other genes and proteins in mammary glands [49].

*PADI2* has been proposed in multiple myeloma as a therapeutic target due to its capacity to remove plasmatic cells’ signaling portion to their abnormal proliferation [50].

The higher PAD4 levels in patients with lung cancer secondary to tobacco smoking suggest that citrullination increase in lung tissue is not necessarily associated with ACPA production; however, it can enhance or be a product of tumor development [46].

Thanks to in vitro studies performed in cancer cells, it has been possible to identify citrullinated proteins like ENO1, HSP60, KRT8, and TUBB, which are citrullinated and expressed in tumors. These proteins affect signaling pathways and cytoskeleton components, but their physiological role changes to an abnormal pathological grade when they are citrullinated [51].

In the following section, we will explore the involvement of PAD proteins in rheumatoid arthritis and lung disease, from protein to genetic alterations, and their impact on these diseases.

### 4.1. Rheumatoid Arthritis

Rheumatoid arthritis (RA) is a chronic autoimmune disease characterized by a symmetric inflammation of peripheral synovial joints [52]. There has been reported an increase in autoantibodies against different biological targets like collagen type II, vimentin, filaggrin, and heat shock proteins, besides highly specific antibodies like anti-citrullinated peptide antibodies (ACPAs), in synovial fluid and serum samples of RA patients [4,53,54,55].

#### 4.1.1. Molecular Mechanisms and Role of PADs in RA

ACPAs are one of the biochemical markers used in the diagnosis of RA, with a sensitivity of >80% and a specificity of 98% in RA patients. However, one of the methods these autoantibodies are produced may be due to the catalytic activity in arginine to citrulline modification (carried out by PAD proteins), which will lead to changes in peptides or proteins that will be found to be citrullinated. These will be the proteins identified by ACPAs [56,57], which can promote the generation of more specific epitopes to ACPAs [58], and PAD2 activity in synovial fluid in this ACPA generation has been described [59]. These processes suggest that RA can be detected in the early stages because of the recognition of peptidyl citrulline epitopes, thus facilitating RA diagnosis [60,61].

Despite the relatively high frequency of inflammatory processes in daily life, only a tiny fraction of the population develops ACPAs, which is highly related to RA development. In synovial fluid samples, PAD enzymes’ presence and activity influence ACPA production [3,59].

The expression of PAD2 and PAD4 is unique to individuals with RA [62]. PAD2 levels are elevated in synovial fluid, along with the presence of ACPA-positive [63].

At the clinical level, antibodies generated by peptide citrullination and antibodies against these proteins (anti-PAD) have been linked to the activity and duration of RA. ACPAs derived from peptide citrullination are associated with RA disease activity, and in carrying anti-PAD (3/4) antibody patients, 53% exhibit low disease activity indices (measured by the CDAI: clinical disease activity index). On average, individuals with ACPAs have a 9.5-year longer duration of RA compared to those with ACPA-negative [64].

Nonetheless, it has been noted that during the pre-diagnostic phase, both anti-PAD4 and anti-CCP antibodies are initially detected at low levels and gradually increase as the disease progresses. These autoantibodies are also linked to erosion and damage that can be visualized through radiographic examination [65]. In early diagnosis RA, PAD levels were elevated and subsequently decreased after treatment with DMARDs (Disease-Modifying Antirheumatic Drugs) [66].

Concerning the PAD2 isoform, it has been established that the presence of anti-PAD2 antibodies is correlated with reduced severity of RA and diminished articular progression, as observed radiographically, regardless of the administration of DMARDs [67]. The presence of these antibodies in RA patients indicates an elevated risk, particularly in cases where multiple autoantibodies are concurrently present [68].

The possible mechanism by which the response to ACPAs increases involves a hapten-carrier model (hapten-citrullinated proteins/-carrier mechanism), with the participation of citrullinated proteins and genetics (*HLA-DRB1* alleles). In this model, peptides are recognized by T-cells, leading to the production of ACPAs [69,70]. However, it is important to note that this mechanism is a hypothesis, and the production mechanism remains unknown.

PAD2, PAD4, and antibody expression have been associated with various proteins mediating inflammation resulting from the autoimmune response. These proteins include TNF-α, where PAD4 may contribute alongside TNF-α in amplifying ACPA production, suggesting a complex positive regulation involving PAD4, citrullinated antigens, ACPAs, and TNF-α, particularly in RA exacerbation [71].

Furthermore, the involvement of cells such as granulocytes and neutrophils has been described at the cellular response level. In the latter, the contribution of PAD enzymes plays an essential role in carrying out chromatin compaction; however, in diseases such as RA, the formation of neutrophil extracellular traps (NETs) has been linked to an inflammatory process [72] where the function of PADs, specifically PAD4, performs hypercitrullination of histone 3. This process is known as NETosis, which is a type of neutrophil cell death, leading to the generation of new antigens that ACPAs and the generation of an inflammatory response can target [73,74].

Among the risk factors described for the development of ACPAs, cultured targets, and increased antigen production are environmental factors such as smoking, occupational exposures, and genetic predisposition, which play an important role in the development and prognosis of RA [75,76,77].

#### 4.1.2. Genetic Associations: Variations in PAD Genes and Their Potential Contribution to RA Susceptibility and Severity

Among the primary risk factors associated with developing this disease and contributing to generating ACPAs, tobacco smoking has been extensively studied. It plays a crucial role in the development of RA by inducing citrullination, a critical mechanism in the development of autoimmune and inflammatory diseases, ultimately leading to an increase in ACPA blood levels [78,79]. As previously mentioned, genetic predisposition factors play an essential role in the development of rheumatoid arthritis. One or more alleles of the shared epitope (SE) of *HLA-DRB1* have been linked to ACPAs and anti-PAD4 antibodies, although other genetic factors can also influence ACPA production. Smokers with RA have shown up to 20 times greater ACPA levels when carrying the *HLA-DRB1*04:01* allele [80,81].

Andrade et al. propose that autocitrullination regulates the production of citrullinated proteins during cellular activation, and polymorphisms influence this process in PAD4, which plays a pivotal role in both its structural configuration and the immune response. They identified multiple citrullination sites in PAD4, including Arg-372 and Arg-374, located within the substrate recognition sites. These sites serve as potential targets for autocitrullination, leading to enzyme inactivation, thereby altering the structure of PAD4 and increasing its recognition by human antibodies. This alteration subsequently affects enzyme–substrate interaction [16,82].

Single-nucleotide variants (SNVs) found in *PADI4* have been identified as being associated with susceptibility to RA. Additionally, interactions between individuals homozygous for the GTG haplotype of *PADI4* and the *HLA*-*DRB1* SE have been linked to the production of anti-CCP antibodies, tobacco smoking, and erosive disease in these patients [83].

*PADI1*, *PADI2*, *PADI4*, and *PADI6* genes have SNVs associated with RA worldwide, as shown in Table 3. Since identification of its role in the cellular nucleus, *PADI4* is the most evaluated gene, mainly in Caucasians (British, North American, Swedish, French, German, and Dutch populations); Asians report associations in Japanese, Chinese, Korean, Indian, and Iranian populations, and in Latin America a single study was carried out in the Mexican population. *PADI2* has had fewer studies associating SNV with RA, only in Chinese, Malayan, and Indian populations. *PADI6* has also had scarce studies, including Chinese and Malayan populations; *PADI1* only has an association report in the Chinese population.

Polymorphisms located in the intergenic region of *PADI3/PADI4*, combined with anti-cyclic citrullinated peptide (anti-CCP) antibodies, have been identified as factors associated with the onset of RA. These genetic variations have been linked to the citrullination process of histones, underscoring their significant role in deimination and subsequent ACPA generation [68]. Citrullination occurring at the Arg-372 and Arg-374 sites can be elucidated as the cause of PAD4 inactivation. This modification preserves the lysine amino acid while removing the guanidine group during citrullination, altering the enzyme’s binding affinity [16].

The SNVs associated with RA and most commonly observed in distinct populations are *PADI2,* particularly the rs1005753 in populations from China, Malaysia, and India, linked to decreased susceptibility to RA (OR < 1). While in *PADI4*, rs1748033, in Japan, Korea, India, and Iran, was linked to an increased RA risk (OR > 1), and rs11203366, in populations from Korea, Germany, France, and Mexico, was linked to susceptibility to RA (OR > 1.5).

### 4.2. Lung Diseases

The involvement of PAD and various citrullinated proteins has been identified in lung conditions, including cystic fibrosis, chronic obstructive pulmonary disease (COPD), idiopathic pulmonary fibrosis (IPF), and autoimmune-associated interstitial disease, such as RA (RA-ILD). These individuals were subjected to biological sample analysis of serum, bronchoalveolar lavage fluids, and sputum, which could reflect the particular damaged site environment (see Table 4).

The participation of PAD has been most frequently reported in interstitial lung diseases (ILD), a condition encompassing numerous subacute and chronic respiratory diseases characterized by a diffuse commitment of lung parenchyma that affects mainly alveolar interstitium and space. According to the ILD classification, we can distinguish primary and secondary types. It has been described that ILD can affect inflammation processes, like fibrosis in the lung parenchyma interstitium [96,97,98,99,100].

Pulmonary damage of collagen vascular diseases can affect almost every region in the lung, including the pleural cavity, alveoli, interstitium, blood vessels, lymphatic tissue, and the upper and lower respiratory tract [101]. Interstitial pneumonia patients associated with vascular collagen diseases have a better prognosis than other ILD patients, considering that collagen is a target for PAD enzymes [102,103,104,105].

#### 4.2.1. Role of PADs in Lung Diseases

Several mechanisms have been proposed to explain autoantibody development and antigen response in the lung and other organs. Paulin et al. propose two possible pathways. First, an immune response against citrullinated peptides starts at joints and spreads to the lungs, resulting in pulmonary interstitium inflammation. Second, usual interstitial pneumonia patients carrying some genetic susceptibility to RA originate an immune response against citrullinated peptides by autoantibodies (ACPAs) in the lung, starting an inflammatory process that affects secondary joints [104,106]. Harlow et al. suggest that the immune response against citrullinated proteins is initiated in the lungs [107].

ACPAs have been strongly associated with smoking and SE in *HLA-DRB1* presence between citrullinated vimentin and α-enolase peptide-1, suggesting that they can trigger an immune response in the lungs that increases citrullinated peptide formation shifting on ACPA response [108].

High levels of protein citrullination and PAD are primarily linked to smoking, particularly PAD2 [109]. Smoking is one of several factors that facilitate the modification of protein citrullination in the lung, potentially resulting in the production of ACPAs and contributing to the onset of autoimmune disease [81] (Table 4).

**Table 4 cells-12-02829-t004:** Association studies of PAD protein in lung diseases.

Protein	Lung Disease	Evaluation	Clinical Finding	References
PAD4	Cystic fibrosis	Autoantibody anti-PAD4 levels	Elevated levels compared with the control group. Negative correlation with pulmonary function.	[110]
PAD4	Cystic fibrosis	Autoantibody anti-PAD4 levels	Increased levels were observed, compared to patients with rheumatoid arthritis. A negative correlation was found between lung function and increased *P. aeruginosa* lung infection.	[111]
PAD2 PAD4	COPD	LL-37 Citrullination	Infiltration of airway cells with PAD4 and Neutrophils. *PADI2* in bronchial epithelial cells and leukocytes. *PADI2* and *PADI4* citrullinated LL-37 at three arginine sites (7, 29, and 34). This led to changes in the production of proinflammatory cytokines, including TNF-α.	[112]
PAD2	IPF	Fibrosis	Citrullinated vimentin in lung macrophages, a significant increase in IPF and IPF-smokers.	[113]
PAD4	IPF RA-ILD	Protein expression in granulocytes and macrophages	Citrullinated peptide increased in IPF and ILD	[114]
PAD4	IPF RA-ILD	Autoantibody production	Anti-PAD4- patients have a higher DTA Fibrosis Score (fibrosis on HRCT was quantified using a data-driven texture analysis DTA fibrosis score). Anti-PAD4+ better lung function (FVC%)	[115]
CEP-1	RA-ILD	Autoantibody production	The presence of anti-CCP/CEP-1+ is associated with ILD and erosive disease.	[116]
ACPA	RA-ILD	Reactivity of peptides Citrullinated proteins	>3-fold increase in reactivity.	[117]
PAD2 PAD4	RA-ILD	Protein levels and SNV in gene *PADI*	Increased protein PAD4 levels in RA-ILD patients and *PADI4* SNV risk genotype carriers.	[118]

Anti-citrullinated alpha-enolase (CEP-1); anti-citrullinated peptide antibody (ACPA); chronic obstructive pulmonary disease (COPD); idiopathic pulmonary fibrosis (IPF); rheumatoid arthritis associated with interstitial lung disease (RA-ILD); high-resolution computed tomography (HRCT); forced vital capacity (FVC); single-nucleotide variants (SNVs).

PAD activity has also been studied in different pulmonary diseases and described in BAL samples retrieved from bronchial mucosa, and biopsies of smokers show an increase in PAD2 expression [81,109]. Table 4 outlines lung diseases in which PAD proteins and citrullinated peptides are linked to pulmonary clinical findings. Regarding cystic fibrosis, PAD4 levels correlate negatively with lung function. Furthermore, the presence of *P. aeruginosa* infection can elevate the protein levels [110,111,119].

In COPD, the citrullination by PAD2 and PAD4 on the anti-microbial, anti-inflammatory peptide LL-37 has been reported. This modification is seen to alter the activity and function of the peptide and proteins signaled during lung inflammation [112].

PAD2 and PAD4 with their substrates (vimentin, fibrinogen, and α-enolase peptide 1) have been found in the lungs, lymph nodes, spleen, and skeletal muscle of COPD patients [108]. Other citrullinated proteins involved in COPD are Fibulin-5 and cytokeratins wich PAD modifies in lung cells and are associated with lung disease in parenchymal destruction processes such as emphysema and COPD exacerbations [120,121,122].

Citrullination and generation of antibodies, such as anti-elastin and anti-CCP, have also been observed in α1-Antitrypsin deficiency. Smoking, one of the main generators of pro-inflammatory molecules, plays an essential role in this process, generating a conducive environment for peptide citrullination in the lung [123,124]. Although smoking is most related to citrullination in pathological processes, exposure to smoke from biomass burning has also been described in individuals without RA in COPD patients [125].

In the context of interstitial lung diseases, an increase in PAD proteins and citrullinated peptides such as vimentin in various cell types has been described in IPF (Table 4). Finally, RA-ILD indicates a potential synergistic interaction between smoking, the elevation of PAD protein levels, and autoantibodies (PAD3/4), which promote an immunological response-favorable environment [118,126].

One of the proteins that may regulate PAD is Gal-9, an immunomodulatory protein that enhances granulocyte activation, resulting in an increase in the expression of PAD4 and the formation of autoantigens [127]. Furthermore, citrullinated proteins have been found to increase in granulocytes and macrophages in IPF and ILD, along with other proteins. Finally, it has been observed that the generation of anti-PAD4 autoantibodies is associated with improved lung function and reduced fibrosis scores in ILD linked to RA [128].

#### 4.2.2. NET-PAD in Lung Diseases

The involvement of PAD4 in the formation of neutrophil extracellular traps (NETs) in the lung is a topic that varies according to the lung disease in which it was described. Physiologically, PAD4 participates in histone citrullination but primarily in infections (*Klebsiella pneumoniae*) [72], which may be found generating new epitopes from the lung; however, not all people with lung disease have such infections.

In cystic fibrosis, there is no difference in NETs and serum-free DNA compared to controls, but increased levels of PAD4, particularly in individuals with *P. aeruginosa* infection, and, in turn, increased levels of ACPAs [111]; however, NETs were not associated. PAD4 was correlated with free DNA (anti-dsDNA IgA autoantibodies) [110].

In COPD, neutrophil response and NET formation have been associated with increased disease progression and decreased lung function [129], but they are also associated with disease severity, not just exacerbations [121].

In fibrotic ILD, the production of NETs in an inflammatory phase has been described, and concerning PAD4, a deficiency of the protein could prevent the development of NETs and, consequently, pulmonary fibrosis, avoiding damage and destruction of the alveolar epithelium and endothelium [130]. It has also been suggested that the NETs levels is related to other proteins, such as the HIF-1α-αMβ2 integrin axis in the pulmonary interstitium [131].

Among the limitations of our review is the lack of genetic information on PAD in lung diseases, especially among RA patients, in which susceptibility has been described in different populations with controvert results, so it is the perspective of the study to evaluate the predisposition of alleles in *PADI* genes and their impact on proteins, as well as involvement in lung diseases. Furthermore, an explanation of how the lung and joint work together in protein citrullination and immune response should be provided.

## 5. Conclusions

The role of peptidyl arginine deiminases in protein citrullination is crucial in physiological pathways. In chronic diseases, environmental factors often lead to continuous alteration of this biological mechanism. In addition, genetic factors play a specific role in their pathogenesis. Genetic variations may represent a new component in the role of PADs in chronic diseases, potentially contributing to developing new pharmacological targets and valuable biomarkers to improve diagnosis and prevention.

In rheumatoid arthritis, single-nucleotide variants in genes encoding PADs and haplotype analyses have been associated with functional changes in the protein and its enzymatic function. Anti-PAD4 autoantibodies and citrullinated proteins are essential in cellular development and physiological and pathological lung maintenance in lung diseases.

## Figures and Tables

**Figure 1 cells-12-02829-f001:**
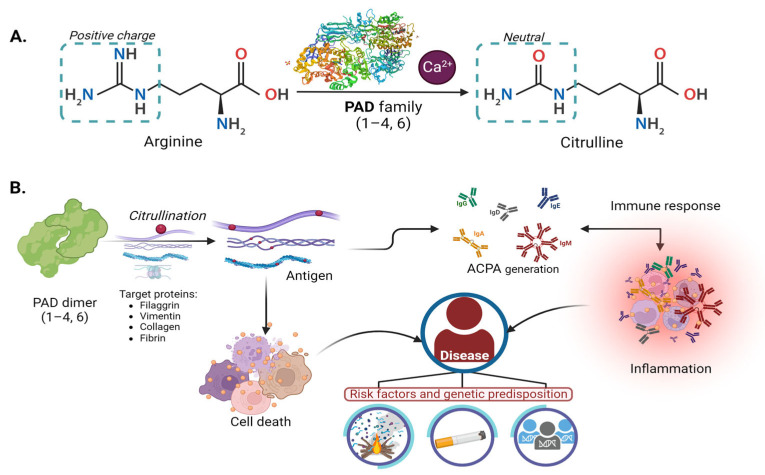
Citrullination molecular mechanism of PAD enzyme physiology and pathology: (**A**) Mechanism of deimination (citrullination) mediated by PAD enzymes dependent on calcium. (**B**) Participation of PAD enzymes in physiopathological mechanisms. The PADs play a role in the citrullination of arginine in proteins such as filaggrin, vimentin, keratin, collagen, and others.

**Figure 2 cells-12-02829-f002:**
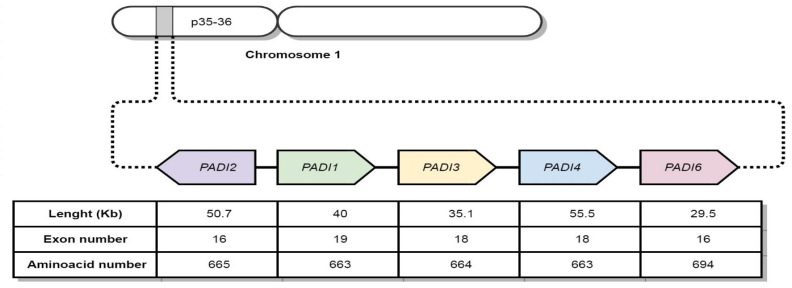
Localization of *PADI* genes encoding PAD enzymes.

**Table 1 cells-12-02829-t001:** Characteristics of peptidyl arginine deiminase (PAD) enzymes.

Enzymes	Intracellular Localization	Main Distribution in Tissues/Organs	Physiological Function	References
PAD1	Cytosol	Dermis and uterus.	Citrullination of simple keratin, differentiation of keratinocytes.	[3,4,5]
PAD2	Cytosol	Skeletal muscle, spleen, brain, salivary glands, uterus.	In the plasticity of the central nervous system. It interacts with the inhibitor of κB kinase, suppresses the activity of NF-κB and stimulation through lipopolysaccharides.	[3,6,7]
PAD3	Cytosol	Hair follicles and epidermis.	Differentiation of epidermis in terminal stages of cells.	[3,6,7,8]
PAD4/5	Nucleus	Hematopoietic system.	p53 regulation and estrogen pathway.	[3,9,10]
PAD6	Cytosol	Ovaries, testes, and peripheral blood leukocytes.	Activation of embryonic genome. Movement (kinesis) of oocytes by microtubules.	[3,8,11,12]

Main characteristics of PAD enzymes including cellular location, tissue distribution, and physiological function. PAD: Peptidyl arginine deiminase enzymes.

**Table 2 cells-12-02829-t002:** Location of *PADI* genes and variants present.

Gene		Variant Total *	Total SNP *n* (%)	SNP				
				srSNP			cSNP	
				3′UTR *n* (%)	5′UTR *n* (%)	Intronic *n* (%)	Synonymous *n* (%)	Non-Synonymous *n* (%)
*PADI1*	*ENSG00000142623*	12,075	11,231 (93.01)	231 (1.91)	23 (0.19)	4927 (40.80)	155 (1.28)	375 (3.11)
*PADI2*	*ENSG00000117115*	21,526	19,288 (89.60)	298 (1.38)	20 (0.09)	10,220 (47.48)	293 (1.36)	539 (2.50)
*PADI3*	*ENSG00000142619*	6150	5638 (91.67)	160 (2.60)	17 (0.28)	3668 (59.64)	151 (2.46)	347 (5.64)
*PADI4*	*ENSG00000159339*	21,122	19,324 (91.49)	93 (0.44)	20 (0.09)	9417 (44.58)	167 (0.79)	421 (1.99)
*PADI6*	*ENSG00000276747*	6998	6214 (88.80)	42 (0.60)	21 (0.30)	4239 (60.57)	200 (2.86)	292 (4.17)

Percentage in parenthesis. * All variants: SNPs, deletions, insertions, among others. srSNP = structural/regulatory SNPs, cSNP = coding SNP. Data taken from Ensembl, August 2023.

**Table 3 cells-12-02829-t003:** Association studies of polymorphisms in *PADI* genes in rheumatoid arthritis.

SNV Tested	Former Nomenclature	Country	*n* (Cases/Controls)	*p*	OR (CI 95%)	Refs.
*PADI1*						
rs2977310		China	429 (266/163)			[84]
*PADI2*						
rs2235926 rs2057094 rs2076616		China	429 (266/163)	<0.01 <0.05 NS	1.70 (1.57–1.86) 1.36 (1.06–1.86) 1.33 (1.00–1.77)	[84]
rs1005753		China	461 (255/206)	<0.05	0.77 (0.51–1.15)	[32]
rs1005753		Malaysia	1502 (516/986)	<0.05	0.97 (0.81–1.17)	
rs1005753		India	664 (379/285)	<0.05	0.78 (0.62–1.00)	
*PADI4*						
rs11203368		China	429 (266/163)	<0.01	0.59 (0.44–0.78)	[84]
rs11203367	*PADI4*_104	Japan	1926 (1019/907)	<0.05	1.14 (1.0–1.29)	[85]
rs874881 rs2240340 rs2240339 rs1748033	*PADI4*_92 *PADI4*_94 *PADI4*_95 *PADI4*_97 *PADI4*_99 *PADI4*_100 *PADI4*_101 *PADI4*_104	Japan	1566 (830/736)	<0.01	1.66 (1.23–2.25) 1.97 (1.44–2.69) 1.89 (1.35–2.66) 1.92 (1.35–2.7) 1.82 (1.33–2.44) 1.75 (1.3–2.38) 1.82 (1.33–2.5) 2.0 (1.41–2.86)	[86]
rs2240340 rs2240337 rs1748033	*PADI4*_94 *PADI4*_102 *PADI4*_104	Japan	2096 (1170/926)	≤0.01	1.23 (1.09–1.39) 1.33 (1.07–1.73) 1.21 (1.07–1.38)	[60]
rs2240340 rs1748033	*PADI4_94* *PADI4_104*	Poland	147(122/25)	NS		[87]
rs11203366 rs11203367 rs874881 rs1748033	*PADI4*_89 *PADI4*_90 *PADI4*_92 *PADI4*_104	Korea	937 (545/392)	<0.05	1.7 (1.3–2.2) 1.7 (1.3–2.2) 1.7 (1.2–2.4) 1.8 (1.2–1.9)	[61]
rs11203366	*PADI4*_89	Korea	133 (50/83)	<0.05	2.22 (0.72–6.86)	[36]
rs1748033	*PADI4*_104	India	151 (95/56)	<0.001	2.27 (1.28–4.011)	[88]
rs11203366 rs11203367 rs874881 rs1748033	*PADI4*_89 *PADI4*_90 *PADI4*_92 *PADI4*_104	UK	1320 (839/481)	NS		[89]
rs874881 rs2240340 rs2240339 rs1748033	*PADI4*_92 *PADI4*_94 *PADI4*_97, 99 *PADI4*_100 *PADI4*_103 *PADI4*_104	UK	222 (111/111)	NS		[90]
rs2240340	*PADI4*_94	Sweden	1031 (1530/881)	NS		[91]
rs2240340	*PADI4*_94	USA	1716 (840/876)	≤0.001	1.24 (1.08–1.42)	[91]
rs11203366 rs11203367 rs2240340	*PADI4*_89 *PADI4*_90 *PADI4*_94	Germany	204 (102/102)	<0.05	1.6 (1.1–2.3) 1.6 (1.1–2.3) 1.6 (1.1–2.3)	[92]
rs11203366 rs11203367	*PADI4*_89 *PADI4*_90	France	680 (405/275)	>0.05 <0.05		[93]
rs1748033	*PADI4*_104	Iran	300 (150/150)	≤0.01	1.63 (1.16–2.29)	[94]
rs1748033	*PADI4*_104	Netherlands	1026 (635/391)	<0.05	1.32 (1.02–1.72)	[85]
rs11203366 rs11203367	*PADI4*_89 *PADI4*_90	Mexico	184 (86/98)	<0.05	2.51 (1.19–5.32) 2.64 (1.21–5.75)	[95]
*PADI6*						
rs10788668 rs2526839 rs6695849 rs7538876		China	429 (266/163)	NS		[84]

OR = odds ratio; 95% CI = 95% confidence interval; NS = not significant.

## Data Availability

Not applicable.

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
