# Peer review of "Peptidyl Arginine Deiminases in Chronic Diseases: A Focus on Rheumatoid Arthritis and Interstitial Lung Disease"

_cells, 2023, doi:10.3390/cells12242829_

Round 1

Reviewer 1 Report

Comments and Suggestions for Authors

Title

Peptidyl Arginine Deiminases in Chronic Diseases: A Focus on Rheumatoid Arthritis and Interstitial Lung Disease

Comments

The topic is interesting. However ,some points need to be addressed

Page

Line

Manuscript

Comments

1

33-43

Protein citrullination is accomplished by a large enzyme family named Peptidyl Ar- 33

ginine Deiminases (PAD), which makes this post-translational modification in a large 34

number of proteins that perform ordinary and pathogenic mechanisms in the body since 35

keratinocyte differentiation to the involvement signaling pathways and/or estrogens reg- 36

ulation, to name a few examples

The introduction is short

Many sentences and paragraphs without references

Research gap should be added before the aim of the review

1

42

Enzymes

What does this title references too ,the enzymes of what ?

2

49

2.2. Classification and Biological Function

The subtitles are not properly arranged as this subtitle under the title of “enzymes “

8

Table 4

ï‚· Infiltration of airway cells with PAD4 and

Neutrophils

No need for these bullets in tables

10

302

Other diseases 302

Multiple sclerosis (MS) is a severe autoimmune and complex

The authors should refer to other diseases according to the tiltle

10

336

citrullinated.[118]

Revise the language writing and grammar and punctuation

10

337

PADI3 is one of the three mutated genes (additionally to TGM3 and TCHH) associ- 337

ated with a rare disease called uncombable hair syndrome [119].

The writing does not cope with the scientific writing. The paragraph is short

350

Add limitations and recommendations

Comments on the Quality of English Language

Moderate editing of English language required

Author Response

Dear Reviewer 

We appreciate your comments and feedback on our manuscript. After a thorough review, we have incorporated your detailed ideas and recommendations to improve areas where our article required clarification. Below, we highlight the changes we have made to address your comments.

Peptidyl Arginine Deiminases in Chronic Diseases: A Focus on Rheumatoid Arthritis and Interstitial Lung Disease

Comments

The topic is interesting. However, some points need to be addressed

  1. The introduction is short. Many sentences and paragraphs without references. Research gap should be added before the aim of the review

A: An improvement was made to the writing of the introduction, in addition to checking the paragraphs without references. A paragraph was added to the introduction to understand the gap between the knowledge and the reason for our study.

  1. What does this title references too, the enzymes of what?

A: The naming of PAD enzymes derives from their various functions within the organism, as demonstrated by the expression of multiple isoforms within this family of proteins. Later discussions regarding genetics spurred the separation of protein and gene levels, leading to this nomenclature.

  1. The subtitles are not properly arranged as this subtitle under the title of “enzymes “

A: Section 2 provides a comprehensive overview of the PAD enzyme family, covering the biological function, mechanism, and regulation of each isoform comprising the family. Subsequently, the genes and implications in various pathologies, focusing on RA and lung diseases, are described. The section's title and subtitle were updated to "2. The PAD Enzyme Family" and "2.2 Biological Function," respectively, to enhance clarity and accuracy.

  1. No need for these bullets in tables

A: Bullets in the table were removed.

  1. The authors should refer to other diseases according to the title

A: A change has been made to the last section to focus on the importance of the study on the involvement of PAD in RA and lung disease.

  1. Revise the language writing and grammar and punctuation

A: The entire manuscript was typed and punctuated.

  1. The writing does not cope with the scientific writing. The paragraph is short

A: It was decided to delete the text of this paragraph due to the lack of information on this isoform in this syndrome, and the section was modified first to mention the diseases globally and then specifically mention the involvement of PADs in rheumatoid arthritis and pulmonary diseases.

  1. Add limitations and recommendations

A: It was added in the last paragraph before the study's limitations and the perspectives to which this manuscript could be applied.

Reviewer 2 Report

Comments and Suggestions for Authors

PAD and protein citrullination play roles in the pathogenesis of many diseases. The authors summarized many studies for this issue.  There are some points need to be addressed in this article. 

1. In table 1, the physiological function and main distribution in tissues/organs of each PAD is too preliminary. Eg. PAD2 did expressed in immune cells. The authors might seem the table 1 in Cell Mol Life Sci. 2022 Jan 25;79(2):94. doi: 10.1007/s00018-022-04126-3.

2. A brief introduction of ACPAs in RA is necessary. 

3.In the section of "Molecular Mechanisms and Role of PADs in RA"

I suggest added the description of NET and the effect of smoking on RA Lung PADs expresssion. 

4. PADs were involved in the pathogenesis of lupus, psoriasis, thrombosis, periodonititis and inflammatory bowel disease. The author should be mentioned about these diseases.

Author Response

Dear Reviewer

We appreciate your comments and feedback on our manuscript. After a thorough review, we have incorporated your detailed ideas and recommendations to improve areas where our article required clarification. Below, we highlight the changes we have made to address your comments.

PAD and protein citrullination play roles in the pathogenesis of many diseases. The authors summarized many studies for this issue.  There are some points need to be addressed in this article.

  1. In table 1, the physiological function and main distribution in tissues/organs of each PAD is too preliminary. Eg. PAD2 did expressed in immune cells. The authors might seem the table 1 in Cell Mol Life Sci. 2022 Jan 25;79(2):94. doi: 10.1007/s00018-022-04126-3.

A: We are looking at the table you mention, but we have chosen only to mention the physiological functions globally, and they are described in detail in the text within the pathologies we focus on, both in RA and lung diseases.

  1. A brief introduction of ACPAs in RA is necessary.

A: It has been added in a small paragraph in section 4.1.1 on the involvement, function, and possible origin of ACPAs in RA because there are several references where they are described as anti-CCP or ACPA, but throughout the text, they are referred to indiscriminately because of the technical reference for the determination of these antibodies (anti-cyclic citrullinated peptide antibodies and antibodies against citrullinated peptides in general).

3.In the section of "Molecular Mechanisms and Role of PADs in RA" I suggest added the description of NET and the effect of smoking on RA Lung PADs expression.

A: As you suggested, NET formation has been included in section 4, but it is summarised in the RA section because NETosis is also a broad mechanism involving PAD enzymes. However, it was a rather exciting topic in lung disease, so it was included in section 4.2.2.

Similarly, smoking is mentioned but is not discussed as an exclusive variable in the development or susceptibility to RA, but is discussed in a general way as an introduction in the form of an environmental exposure factor for the introduction of genetic predisposition.

  1. PADs were involved in the pathogenesis of lupus, psoriasis, thrombosis, periodontitis and inflammatory bowel disease. The author should be mentioned about these diseases.

A: They are mentioned at the beginning of section 4 but have been grouped under chronic and autoimmune diseases heading. “It has been related to some chronic and autoimmune diseases, including rheumatoid arthritis (RA)”

Round 2

Reviewer 2 Report

Comments and Suggestions for Authors

The authors have answer my questions.